# Screen Printing Carbon Nanotubes Textiles Antennas for Smart Wearables

**DOI:** 10.3390/s21144934

**Published:** 2021-07-20

**Authors:** Isidoro Ibanez Labiano, Dilan Arslan, Elif Ozden Yenigun, Amir Asadi, Hulya Cebeci, Akram Alomainy

**Affiliations:** 1Department of EECS, Queen Mary University of London, London E1 4NS, UK; i.ibanezlabiano@qmul.ac.uk (I.I.L.); a.alomainy@qmul.ac.uk (A.A.); 2Aerospace Research Center, Istanbul Technical University, 34469 Istanbul, Turkey; arrslan.dilan@gmail.com (D.A.); hulya.cebeci@itu.edu.tr (H.C.); 3School of Design, Textiles, Royal College of Art, London SW7 2EU, UK; 4Department of Engineering Technology & Industrial Distribution, Texas A&M University, College Station, TX 77843-3367, USA; amir.asadi@tamu.edu; 5Faculty of Aeronautics and Astronautics, Istanbul Technical University, 34467 Istanbul, Turkey

**Keywords:** e-textiles, wearables, screen printing, flexible printed antennas, carbon nanotubes inks

## Abstract

Electronic textiles have become a dynamic research field in recent decades, attracting attention to smart wearables to develop and integrate electronic devices onto clothing. Combining traditional screen-printing techniques with novel nanocarbon-based inks offers seamless integration of flexible and conformal antenna patterns onto fabric substrates with a minimum weight penalty and haptic disruption. In this study, two different fabric-based antenna designs called PICA and LOOP were fabricated through a scalable screen-printing process by tuning the conductive ink formulations accompanied by cellulose nanocrystals. The printing process was controlled and monitored by revealing the relationship between the textiles’ nature and conducting nano-ink. The fabric prototypes were tested in dynamic environments mimicking complex real-life situations, such as being in proximity to a human body, and being affected by wrinkling, bending, and fabric care such as washing or ironing. Both computational and experimental on-and-off-body antenna gain results acknowledged the potential of tunable material systems complimenting traditional printing techniques for smart sensing technology as a plausible pathway for future wearables.

## 1. Introduction

Electronic textiles (e-textiles) that involve the combination of electronics and textiles introduce “smart” functions to textile products [1]. Electronic and sensory functions on the flexible substrates, including fabrics, have been demanded in many wearables and Internet of Things (IoT) applications for wireless sensor networks (WSN) with a strong acceleration embracing sensory features [2,3]. In the context of wearable communication, antennas are a key component that adheres to specific requirements, including flexibility, conformality, being low profile, and lightweight [4]. Unlike conventional antennas, wearable antennas should be analysed in dynamic environments to simulate complex real-life situations, such as being in the proximity of a human body and under deformation of wrinkling, bending, or cyclic deformation such as washing and ironing. In the emerging field of flexible antennas, either knitted or woven, different textile forms can withstand bending, twisting, and stretching and promise longevity in critical components. Add-on textile processes, including printing [5,6] and lamination, can induce innovative surface features and layers using conductive materials [7]. However, the success of final executions on the textiles is limited by the compatibility of ink and intrinsic properties of a conductive material such as conductivity, durability, susceptibility of washing, and humidity. Until now, various kinds of conductive inks with different fillers such as metal nanoparticles [8,9,10,11], polymers [12], carbon nanotubes [13], and/or organic metal complexes [14] have been developed for forming conductive patterns. In the aim of scalable processes for printed electronics, inkjet and screen printing techniques tuning the liquid phase inks with high efficiency, stability, low cost, and zero waste materials have been employed [15,16,17]. A recent technology advancement attracting much attention from the design and research community is inkjet printing. Despite its widespread use, inkjet printing has drawbacks and difficulties associated with ink stability, clogging of nozzles, and surface limitations, including rough and porous textile surfaces due to its low viscosity and long printing times [18,19]. 

Screen printing is the most common and straightforward conventional printing method, being relatively effortless, versatile, affordable, fast, and adaptable [15,18,19,20]. Screen printing is a making method that allows transferring the ink through a stencil design to a flat surface by using a squeegee and mesh screen usually made from silk or nylon. Several parameters are critical in producing high-quality patterns, such as squeegee shape, speed and pressure, snap-off distance (a gap between the screen and substrate), and mask clearance [21,22]. The formulated inks for screen printing are qualified with rheology analysis such as shear viscosity as a function of shear rate [15]. During printing, the ink with high viscosity should be deposited slower with the squeegee to allow sufficient deposition time for the ink to flow through the screen meshes. When the ink has low viscosity, the speed should be higher to prevent uncontrollable spreading [23]. Tseng et al. examined the rheological behaviours of nickel nanoparticles dispersed solvents, using various organic surfactants over a shear-rate range 10^0^–10^3^ s^−1^. The results demonstrated that nanoparticle viscosity reduced as much as 40–70% with the surfactants, and pseudoplastic flow behaviour was observed at the shear-rate regime often encountered in most screen printing processes [24]. 

Hong et al. demonstrated UV-curing conductive inks with 60 wt.% silver nanoflakes for screen printing on standard fabrics. The UV-curable conductive ink contains 24 wt.% polymers, and 10.8 wt.% diluents showed a viscosity of 176.29 Pa·s at a shear rate of 0.1 s^−1^. Ultra-high frequency radio-frequency identification (UHF RFID) tags were manufactured using this UV-curable conductive ink by screen printing with a conductivity of 6.02 × 10^6^ Sm^–1^ [25]. Despite reported promising performances, the metallic conductive inks cannot fully meet the haptic requirement of textiles since they become gradually oxidised and lose performance, bringing additional weight and thickness to the structure that disrupts comfort. Thus, the researchers have alternative material systems to fulfil textiles sensory values while adding electronic functions. In an attempt to create flexible sensors and antennas, different sp^2^ carbon assemblies, such as graphene and carbon nanotubes (CNTs), grasp the best compromise between features such as high electrical conductivity, high thermal stability, strong mechanical properties, high corrosion resistance, and low density [26,27]. CNTs with their unique characteristics, such as high intrinsic current mobility, electrical and thermal conductivity, mechanical stability, and low-cost production capabilities are of interest for electronic and bio applications [28,29,30,31,32,33].

Among non-metallic conductive materials, CNTs play a pivotal role, owing to the aforementioned characteristics that make them strong candidates for flexible devices, thus attracting attention in wearable antennas and sensors [34,35,36]. CNTs have been previously repurposed for wireless communication [37], including on-body applications [38], and layered up in the form of thin conductive films [39]. However, the challenges in scaling up processes lead the researchers to repurpose conventional textile printing techniques for mass production that can boost the manufacturing of e-textiles. Previous works reported CNTs laminated antennas on flexible substrates such as Kapton [37], paper [40], and polystyrene, whereas the contribution of fabric as a substrate was not navigated for microwave antennas, near-field (NF) communication, and RFID [41,42]. In printed electronics, the conductive inks containing binders serve to secure together the other components, such as granular powders or conductive fillers, to create a smooth and homogenous film deposited on top of a substrate. However, these binders need to be cured or fixed through high-temperature processes, such as crosslinking, annealing, and sintering. These post-processes require precision not to induce deviation in material qualities due to applied heat, impacting final antenna performance [43]. Furthermore, the heat processing steps are not incompatible with other flexible substrates such as papers and natural textile materials. Therefore, carbon-based ink formulation requires understanding the material to develop binder-free or low-temperature processing solutions while retaining high conductivity.

In an attempt to design sensory fabrics by CNTs, [44] screen printed fabrics could be created by incorporating aqueous CNTs ink. The results showed that in the first layer of deposition, the CNTs based fabric surface sheet resistance was 141.5 Ω/sq, then it was decreased to 50.75 Ω/sq by applying three layers. Despite the promising electronic function for wearable applications, the poor solubility of CNTs in water due to their hydrophobic surfaces and highly attractive inter-van der Waals interactions between CNTs makes the process under control [45]. Menon et al. proposed a room temperature curable conducting ink using multiwalled carbon nanotubes (MWCNTs) for a printable electronic application [46]. In another study, flexible substrates such as Mylar^®^, silicone rubber, and photo paper were explored as challenging with screen-printing. As a result of using 9 wt.% MWCNTs, the sheet resistance was reported in the range of 0.5–13 Ω/sq, whereas the use of a binder in their suspensions disrupted the electrical conduction network [47]. Despite the strong evidence of CNTs based inks in printable electronics applications, the paucity of existing data for developing water-based and binder-free inks could discourage researchers from designing their ink formulations. 

In an attempt to rediscover earth materials in electronic functions, carbon nanomaterials still require support to bring their superiority forward for scaled-up processes. Hence, the researchers have started seeking out nature-derived-non-petroleum binders to promote interfacial bonding. Among a broad range of biopolymers, cellulose, an abundant biopolymer, promises favourable properties such as low cost, biodegradability, biocompatibility, nontoxicity, and mechanical strength [48]. Cellulose nanocrystals (CNC) are extracted from natural cellulosic materials by controlled acid hydrolysis, in which surface sulphate groups enable excellent colloidal stability [49]. It has polar and nonpolar groups that could assist dispersion in both apolar and polar solvents. A deeper understanding of intermolecular interactions in ink formulations could avoid phase separation in large-scale processes. For instance, the nonpolar areas of cellulose nanocrystal can interact with hydrophobic CNTs and aid in bridging between two nanotubes [50], and highly stable CNTs dispersions were possible by wisely introducing CNC [51]. This study investigated the CNC-assisted MWCNTs and single-walled carbon nanotubes (SWCNTs) aqueous suspensions’ stability and dispersing capacity. MWCNTs/CNC aqueous suspension achieved its maximum dispersion yield of 60% at the MWCNTs concentration of 0.05 wt.%.

In comparison, SWCNTs/CNC aqueous suspension reached its maximum dispersion yield of 80% at the SWCNTs concentration of 0.01 wt.% In another study, the optimal parameters of the CNTs/CNC dispersion for both SWCNTs and MWCNTs have been revealed [45]. Respectively, in SWCNTs/CNC (0.55/10 g/L: sonication time of 2.5 hours) and MWCNTs/CNC (0.33/10 g/L: sonication time of 1 hour) dispersions, the yield was reported as high as 72% and 80% with a power density of 0.7 W/mL.

Previous works reported isolated challenges regarding using carbon-based inks, screen-printed techniques, and flexible substrates for on-body applications. This study attempts to develop flexible and lightweight CNTs based textile antennas that can extend the possibilities of current state-of-the-art wireless body-centric systems by promising a new sustainable material medium for e-textiles. The research embraces a holistic approach from behavioural decisions and understanding of the wearer’s priorities to environmental and economic concerns related to the textile screen printing processes. In each intervention point, we elaborated on the material and process requirements to offer a working communication channel in the range of microwave frequencies. The previous section of the article presents a complete introduction to the topic. Section 2 will provide information regarding the materials and methods used to fabricate the prototypes. At the same time, both numerical and experimental analyses have been carried out in Section 3, closing with conclusions in Section 4.

## 2. Materials and Methods

### 2.1. Materials

CNTs were purchased from Sigma Aldrich (Sigma Aldrich, SouthWest NanoTechnologies, Norman, Oklahoma, US) having a purity of ≥98%, an outer diameter: 10 nm, and length: 3–6 µm. CNCs were provided from CelluForce, Windsor, QC, Canada, with a diameter of 2.3–4.5 nm and an average length of 44–108 nm [52] and were mixed with deionized water (DI-H_2_O). Screen-printing was performed using a polyurethane squeegee and 26 × 32 cm wooden silk frame with mesh size 55 thread/cm (55T) and 90 thread/cm (90T). Two different woven fabrics of 100% cotton (CO) and 65% cotton–35% polyester (CO–PES) were used as the textile substrates for screen-printing of CNTs/CNC inks. The fabric patterns were plain weave for both CO and CO–PES. The mass of the fabrics was 116 g/sqm and 122 g/sqm for CO and CO–PES fabrics, respectively. All chemicals were used as received.

### 2.2. Preparation of CNTs/CNC Ink Dispersions

The CNTs and CNC dispersions were prepared by mixing CNC with DI-H_2_O at a weight ratio (wt.%) of 0.5, 1, and 1.5%. The CNC/DI-H_2_O dispersions were sonicated with a microtip sonication (Sonics VCX750) for 35 min at 20 kHz, 8 Watt, and 20% amplitude at 480 J/min. Then, CNTs were mixed with CNC aqueous dispersion to derive the ink composition, and this mixture will be named CNTs/CNC ink throughout this research paper. The CNTs/CNC ink formulations were prepared with a weight ratio of 1:1 (for 0.5, 1, 1.5 wt.% of CNC) and 1:2 (1 wt.% of CNC). CNTs/CNC inks were dispersed using a microtip sonicator for 1 h at the same aforementioned sonication process. No conventional dispersants and surfactants were employed for preparing CNTs/CNC inks, and all inks were used directly without any post-treatment. The details of CNTs/CNC ink formulations related to sample codes were given in Table 1. Different concentrations were used for both planar inverted cone antennas (PICA) and LOOP designed antenna models, detailed in the antenna design section.

During CNTs/CNC ink preparation, when the CNTs were at higher concentrations, increased sonication energy and longer time was applied for an efficient dispersion [53]. Thus, the sonication time was increased to 1.5 hours for CNTs/CNC—1.5:1.5 compared to 1-hour sonication for all other compositions while keeping the frequency and amplitude the same. During sonication, the dispersion container was always ice-cooled to prevent excess heating resulting in instability of viscosity.

### 2.3. Antenna Designs

In this section, the two different antenna designs named planar inverted cone antenna (PICA) and LOOP are presented. Among two different designs, ultra-wideband (UWB) PICA antenna was selected due to the ability to provide appealing features such as high capacity with significant bandwidth signals, multi-path robustness, and low-power requirement. UWB antennas have been extensively studied in wearable and healthcare applications. Although this piece of research does not cover time domain characterisation (group delay, normalized amplitude, fidelity factor, or power spectral density), it provides enough evidence in terms of antenna performance when placed on/off body settings. LOOP antenna (also called a magnetic loop) promises inherent robustness when in proximity to the human body due to its sensitivity to the magnetic field where the human body has an important impact on the electric field. These two designs reveal different communication channels for textile-based antennas at highly dispersive mediums such as the body. For each design, two prototypes with varying formulations of ink were fabricated by screen printing of five layers onto the cotton derivative substrates.

#### 2.3.1. PICA

The first antenna model analysed consists of a UWB, with a radiation quarter-wavelength (*λ*/4) monopole in the shape of a PICA and a coplanar waveguide (CPW) as the feeding technique. The antenna geometry was determined based on the monopole disc principles [54], thus an optimization process was completed to decide the antenna dimensions as depicted in Figure 1. UWB designs are used in different sensing applications like temperature, moisture, strain, microwave imaging, etc. [55]. The proposed antenna has been proven to be a solid alternative to the off-the-shelves solutions for wearable devices. The textile feature offers structural advantages, such as lightness, washability, and drapability [7].

#### 2.3.2. LOOP

A wearable loop antenna structure is printed on a very thin non-grounded 0.245 mm-thick textile cotton fabric, where the loop antenna is of size *L* × *W* = 50 mm × 35 mm, as depicted in Figure 2, designed to resonate around 2.45 GHz (industrial, scientific and medical, ISM band), to meet the criteria of sensing and healthcare applications. The actual loop is *Lp* × *Wp* = 43.5 mm × 25.1 mm, respectively, and the gap at the feeding point is 1 mm. The printed trace width of 1 mm and 2 mm, previously studied, showed a stronger resonance frequency for 2 mm; thus, this width was chosen in this study [56].

### 2.4. Screen-Printing of PICA and LOOP Antennas onto Textiles

A manual screen-printing setup was used to prepare the CNTs/CNC patterns of PICA and LOOP antenna designs on the woven fabrics of CO and CO–PES. First, woven fabrics were cut by referring to computational models described in Section 2.3 and were conditioned at 85 °C for 24 hours. All fabrics were weighed until achieving a constant weight before the screen-printing process to record the baseline weight of the unprinted fabric. Conductive CNTs/CNC ink was printed onto woven fabrics in a layer-by-layer fashion and presented schematically in Figure 3. After each printing layer, all fabrics were weighed to measure the total ink penetration for identifying the related thickness through the area and density calculations [57].

#### 2.4.1. Printing PICA Textile Antennas

Conductive inks of CNTs/CNC—0.5:0.5, CNTs/CNC—1:1, and CNTs/CNC——1.5:1.5 were prepared initially as described earlier onto CO fabric only. To obtain comparable results, each sample and layer were screen printed under the same conditions. First, 0.5 mL of ink was dropped into the screen printing frame and by using the squeegee, and 5 times of layering were performed to achieve a uniform and homogeneous printing pattern onto CO fabric. The structure was held 30 seconds on the CO fabric before being lifted off and then cleaned with distilled water to prevent pores from clogging. Afterwards, the printed pattern was dried at 85 °C in the oven until the moisture was removed, then weighed. Under the same printing conditions, five layers of printing were deposited on the CO fabric using both formulated inks. Still, after the first layer, the amount of ink and the number of back-and-forth movements of the squeegee were gradually reduced to prevent over-spreading inks on the fabric. The details of printing process parameters are presented in Table 2 for the number of layers, dispersion volume, and squeegee movements. To achieve a homogeneous printing pattern and finely tuned sharp corners and lines, a systematic squeegee movement is also controlled with the number of movements and correlated with the weight of the fabric as well. The weight of the printed fabrics was measured for every print layer to control the amount of total CNTs/CNC coating precisely. The PICA designed antenna prototypes with five layers of thickness were 25 ± 2.9 μm for CNTs/CNC—1:1 and 35 ± 2.1 μm for CNTs/CNC——1.5:1.5, respectively. Due to the non-homogeneous distribution of the ink, CNTs/CNC—0.5:0.5 composition did not result in a well-defined printing pattern for PICA antenna, which was attributed to the viscosity of the ink and higher water absorbance of CO fabric.

#### 2.4.2. Printing LOOP Textile Antennas

The LOOP antenna designs were prepared with similar process parameters and steps as explained in the printing PICA-designed textile antenna section. The LOOP antennas were printed onto CO and CO–PES fabrics. Within the printing experiences observed in PICA antennas onto CO fabrics, CO–PES as an alternative textile substrate with lower water absorbance was also used for LOOP antennas. CNTs/CNC inks with weight ratios of 1:1 and 0.5:1 were utilized to prepare LOOP model samples. Similar printing protocols for PICA-designed antennas were used to print LOOP antennas. The details of the number of layers, the volume of dispersions, and squeegee movement is presented in Table 3. The volume of distribution was lowered gradually from the first to fifth layers to avoid the excess use of CNTs/CNC ink. The final prototypes had an average thickness of 14 ± 5.2 μm for CNTs/CNC—1:1 and 6.4 ± 0.7 μm for CNTs/CNC—0.5:1 printed LOOP antennas, respectively. 

### 2.5. Rheological Behaviours

Rheological measurements of CNTs/CNC-based inks were conducted using an rotational rheometer (TA Instruments, Discovery Hybrid Rheometer 2) equipped with a plate–plate geometry diameter of 25 mm. Measures of 0.5 mL volume inks with 25 mm diameter and 1 mm gap thickness were injected between aluminium plates and tested at 25 °C. First, strain sweep tests were performed to determine the linear viscoelastic region (LVR) and then the frequency sweep tests were carried out using a strain amplitude within the linear viscoelastic regime. The complex viscosity was recorded as a function of angular frequency between 0.1 and 10^2^ rad/s.

### 2.6. Electrical Conductivity Measurements

The electrical conductivities of conductive surfaces were measured by a Four-Point Probe (FPP 470, Eltek Electronics, Istanbul, Turkey) at room temperature. For each sample, at least seven measurements were taken by varying sites on the surface; the mean conductivities are reported in Section 3.3.

### 2.7. Fabric Care: Washing, Ironing, and Comfort

A continuous washing cycle was performed to evaluate the adhesion of CNTs printed layers onto only CO fabric with PICA antennas using the revised protocol based on BS EN ISO 6330 via a domestic washing machine (D1 6101 ES A, BEKO, Istanbul, Turkey) due to the lack of a standard protocol in assessing the wash durability of e-textiles. In this study, only CO fabrics were evaluated in terms of care performance. Both CNTs/CNC—1:1 and CNTs/CNC—1.5:1.5 printed onto CO fabrics were washed and ironed. The wash process was performed in a hand wash machine at 30 ± 3 °C using a detergent without an optical brightener and sulphate. Throughout the washing, the textile samples underwent an initial cold wash, three rinsing cycles, two draining cycles, and a dry spin (400 speed). Each printed fabric was washed four times. In between cycles, the surface resistance change of PICA antennas was measured before and after washing. According to the BS EN ISO 105-X11:1996 standard, ironing was then applied, and the resistance was measured after each washing/ironing cycle as described in Section 3.4.

The weight penalty is one of the significant issues within wearable systems in so much as it has a negative impact on the final user’s experience. CNTs antenna save the lightweight feature in comparison with more dense materials such as traditional bulky metals. Similarly, other fabrication methods such as lamination use extra layers for bonding, adding a significant proportion to the final weight of the antenna. The CNTs PICA antenna fabricated has attained a total weight of 53 g, ~12% lighter than the previous design of multi-layer graphene and ~29% less weight with respect to a copper tape model of 75 g [7].

### 2.8. Dielectric Characterization of the Fabric Substrate

Dielectric properties of the substrate fabrics were characterized to carry out further numerical analyses. Two parameters, such as the dielectric constant (*ε_r_*) and the dissipation factor (*DF* or *tanδ*), were measured to represent CO fabric’s dielectric characteristics; a resonant material characterization technique was employed [58,59]. A cavity perturbation method, by means of a split cylinder resonator for material characterization (Agilent 85072A), working at 10 GHz, was performed by using the Keysight material characterization software (N1500A-003 Materials Measurement Suite 2015). The textile sample was placed in the aperture of the cavity and measured, then rotated 90 degrees and measured again in the weft direction. For 100% cotton plain weave fabric, a dielectric constant of 1.58 and a dissipation factor of 0.02 were calculated, and for the CO–PES fabric, 1.62 *ε_r_* and 0.018 *DF* was noted.

### 2.9. Antenna Measurement Considerations

The fabric specimens were measured using a vector network analyser (VNA), PNA-L Agilent N5230C, and inside a mobile antenna electromagnetic compatibility (EMC)-screened anechoic chamber to examine the far-field properties radiation patterns of the antenna under test (AUT). The chamber is fitted with two open boundary quad-ridge horn antennas operating from 400 MHz to 6 GHz (ETS-Lindgren 3164-06) and from 0.8 to 12 GHz (Satimo QH800) allowing vertical and horizontal linear polarization measurements [60]. For wearable sensing applications, the performance of a wearable antenna when in proximity to the human body needs to be examined. During on-body numerical simulations, all electrical properties (*ε_r_**, tanδ*, conductivity (*σ*), and resistivity (*ρ*)) of human tissues, comprising: dry skin, fat, and muscle have been considered from the available libraries in the CST microwave studio and literature [61], Table 4. 

The antenna under test (AUT) was carried out by placing the device on the phantom model. A human torso phantom was filled with a solution that mimics the electromagnetics properties [7] of the dissipative human body as shown in Figure 4a. In this study, the solution consisted of 79.7% of deionized 0.25% of water sodium chloride, 16% of Triton X-100 (polyethylene glycol mono phenyl ether), 4% of diethylene glycol butyl ether (DGBE), 0.05% of boric acid, and was filled to the phantom in Figure 4a. A 3D view of a numerical phantom model to characterize the performance of wearable antennas within proximity to a human body is sketched in Figure 4b and the model consists of a three-layer block of 1 mm of dry skin, 3 mm of fat, and 40 mm muscle, using the data listed in Table 4.

The radiation efficiency of an antenna (*η*) is explained as the fraction of the radiated power *P_rad_* between the input power *P_in_*: *η* = *P_rad_/P_in_* = *P_rad_/P_rad_* + *P_loss_* = *R_rad_/R_rad_* + *R_loss_*,(1)

*P_loss_* is the lost power, and *R_rad_* and *R_loss_* are the radiated and loss resistance. Different methods could be used for measuring *η*, depending on the antenna design and size in the study; the cylindrical wheeler cap technique was used [62,63,64], as displayed in Figure 5 by following the below equations for different antenna lengths.

Short antennas (*L* ≤ *λ*/10, with *L* the antenna’s size and *λ* the operational wavelength):*η* = [*S*_11*wc*_]^2^ − [*S*_11*fs*_]^2^/1 − [*S*_11*fs*_]^2^,(2)

Moderate length antennas:*η* = 1 − ((1 − *S*_11*fs*_)(1 + *S*_11*wc*_)/(1 + *S*_11*fs*_)(1 − *S*_11*wc*_)),(3)

The radiation resistance is zero with the cap on, and the antenna reflection coefficient was measured and referred to as *S*_11*wc*_. With the cap off the radiation resistance is that of the antenna radiating into free space, and the antenna reflection coefficient was measured and referred to as *S*_11*fs*_. 

## 3. Results and Discussions

### 3.1. Stability of Printing CNTs/CNC Dispersions

Shelf life of CNTs-based inks determines the practical use in large scale applications such as electronic and smart wearables. The main concern for carbon-based inks is the difficulty in maintaining stability of the suspension and agglomeration of particles through time. Several strategies have been employed to increase the stability of the inks, such as introducing functional solvents to maintain effective and constant dispersion [65], applying a powerful sonication process [53], and/or adding dispersing and stabilizing agents [66] as copolymers. When screen-printing is in place for creating sensory surfaces, the particle formation, clogging of meshes, and the aggregation of inks are major process-related challenges to achieve successful patterns. Efficient and homogeneous dispersion of ink that can overcome the stability issues for inks could result in successful printing and improved shelf life. CNTs tend to agglomerate in aqueous solutions due to their high surface energy and inadequate chemical affinity with the dispersing medium; thus, preventing the formation of large bundles is difficult to control. In this work, the colloidal stability of CNTs/CNC with a weight ratio of 1:1, 1.5:1.5, and 0.5:1 was investigated by sedimentation tests, and photos of dispersions from 1 hour to 1 week (left to right)were presented in Figure 6. Additionally, the CNTs dispersion stability in DI-H_2_O was also explored and compared with different compositions of CNTs/CNC inks. It is evident that CNTs/CNC exhibited excellent stability even after a week, whereas the pristine CNTs have poor dispersion stability in DI-H_2_O. Although not presented here, after several months, these inks were still in a good stable condition without any sedimentation. This was attributed to the fact that the -OH groups of CNC, creating the hydrogen bonding with water molecules, enabled the CNC to be effectively dispersed in an aqueous medium. Peculiarly, sulfuric acid-treated CNC presented here has negatively charged sulphate half-ester groups and promoted colloidal stability [67]. Additionally, the amphiphilic nature of CNC with the nonpolar groups had an affinity to interact with hydrophobic CNTs and finally enabled the bridging in between nanotubes to create a network. A similar observation was performed by Asadi et al. for CNTs/CNC dispersions with a weight ratio of 1:1, 1:2, and 1:4 maintaining their colloidal stability up to six months for reinforcing the carbon-fibre-reinforced-polymer composites [52].

Many carbon-based inks were employed by hazardous solvents that were hard to manage for environmental concerns. The most prominent and widely used solvents were toxic as N, N-dimethylformamide (DMF) [68], absorbable through the skin as dimethyl sulfoxide (DMSO) [69], and carcinogenic as N-methyl-2-pyrrolidone (NMP) [70]. During the processing of solvent-based ink, the evaporation of these compounds causes a diversity of health, safety, and air contamination problems. However, the CNTs/CNC inks referred to in this study were promoted to be one of the very few aqueous ink compositions without any surfactant or dispersants and can be referred to as environmentally-friendly for the printing industry.

### 3.2. Rheology of CNTs/CNC Ink and the Quality of Printed Patterns

For a successful print pattern created using screen printing, especially when thinner lines and sharper edges were in place, the rheology of the ink plays a critical role. Many factors such as the solvent system and solid content influenced the dispersion degree of particles in ink, resulting in different rheological behaviours. Nowadays, inkjet printing is one of the methods widely used in printing electronic circuitry. The rheology of the designed ink has to be formulated to avoid nozzle clogging with finely tuned particle sizes. Thus, the range of printable viscosities was restricted to 1–10 mPa·s and the particle size must be smaller than the nozzle output diameter [71]. On that note, these restrictions require low solid content and finely tuned particle sizes which may result in limited material availability. Additionally, with their scalability and adequate printing times compared to inkjet printing, screen-printing could lead the way towards printed smart wearables. Since high solid content is correlated with the increased electrical conductivity, the ink formulation with high solid content reached to an elevated viscosity regime of 10^2^ to 10^6^ mPa·s [71]. The CNTs/CNC ink compositions with varying weight ratios were studied by monitoring their viscosities by a plate rheometer and it was reported how the CNTs’ content and the properties of CNC influenced the rheological profile of corresponding inks. Flow behaviour of CNTs/CNC inks with different CNTs and CNC concentrations were tested at ambient temperature. Figure 7 showed the complex viscosity as a function of frequency for CNTs/CNC—0.5:0.5, CNTs/CNC—1:1, CNTs/CNC—1.5:1.5, and CNTs/CNC—0.5:1 inks as listed in Table 5.

All inks exhibited a shear thinning behaviour at a frequency regime of 10^−1^ to 10^2^ rad/s. Hence, higher CNTs concentrations caused a viscosity increase which was attributed to the enhanced Van der Waals forces between the nanotubes [72]. All ink compositions showed a gradual decrease in complex viscosity since shear forces assisted the breaking of CNTs clusters and aligned particles along the flow direction; consequently, sliding of smaller CNTs clusters and a shear-thinning profile could be observed [73,74,75]. As a critical requirement, although screen-printing inks had a higher viscosity regime compared to inkjet printing inks, still the viscosity should be low enough to enable the ink to pass through the screen meshes with the squeegee but high enough to allow the conductive ink to cover the overall pattern [76]. The variables of the printing process such as mesh size, squeegee movement should be well correlated with the complex viscosity of the CNTs/CNC formulated inks that cause rheological changes during the flow. In manual processes like screen printing in this study, the control on the ink injection rate is not well-controlled to match with complex viscosity ranges. Nevertheless, the physical characteristics of screen-printed patterns were also observed to assess the quality of printing as a measure. Several trials by tuning different ink compositions and mesh sizes of the print frame were conducted to achieve sharp and clean corners and lines onto rough surfaces. CO and CO–PES fabrics were studied in terms of water affinity, where CO–PES fabrics presented a lower water absorbance than CO fabric. Initially, the mesh size of the print frames such as 55T and 90T was selected by keeping mind of the lowest viscosity ink composition as CNTs/CNC—0.5:0.5. As depicted in Appendix A, for both PICA and LOOP designs, 55T frames, which have larger pore sizes, were not compatible with the low viscosity (47.3 ± 0.2 Pa·s at 0.1 rad/s) inks. 

Additionally, the high water affinity of CO fabrics also caused the spread of the ink, particularly around sharp corners and vertical lines, and damaged the quality. Appendix A shows that the screen printing success on CO fabric was improved with very few scattered regions on the prints using a 90T frame in a circular pattern. However, the sheet resistance of the CNTs/CNC—0.5:0.5 pattern was very high as 75.5 Ω/sq after five layers and concluded as insufficient for an antenna application. Afterward, in an attempt to achieve a higher conductivity, CNTs/CNC—1:1 composition was used to explore the higher concentration of CNTs on the overall resistance of the printed patterns.

The screen printing process was performed by a 55T frame based on the feedback of early investigations since a larger pore size would help more particles to accumulate and adhere onto the fabric surface effectively. Hence, CNTs/CNC—1:1 sheet resistance was measured as 45 Ω/sq after five layers without any loss in the print quality, as shown in Appendix A. As a result, 55T was chosen as the optimal mesh size for formulated CNTs/CNC ink compositions. For an effective antenna performance, the sheet resistance of patterned surfaces should be lower than the reported values. On that note, CNTs content was also increased in ink formulations, and the resistance change was monitored on printed PICA and LOOP antenna patterns by CNTs/CNC—1:1 and CNTs/CNC—1.5:1.5. When the CNTs concentration was increased up to 1.5 wt.%, the pores of the meshes were relatively clogged, as presented in Figure 8b due to higher solid content than 1:1 ink composition. However, for both of the 1:1 and 1.5:1.5 ink compositions, the overall quality of the printed pattern was good, as shown in Figure 8a, even in fine details such as sharp corners and clean lines on CO fabric. When the ink viscosity was tuned by increasing the CNTs up to 1 wt.%, promising print quality was achieved on CO fabric, as illustrated in Figure 8c.

Additionally, the CNTs/CNC—1:1 had a viscosity of 141 ± 6.8 Pa·s at 0.1 rad/s, showed adequate shear flow properties for screen printing application with comparable viscosities from others [76,77,78]. For CNTs/CNC—0.5:1, the viscosity was 81.6 ± 6.9 Pa·s at 0.1 rad/s exhibiting similar flow properties with CNTs/CNC—0.5:0.5 as well. In this viscosity regime, CO–PES fabrics were preferred as the textile substrates with lower water absorbency rather than employing CO fabric. Even though there were small casualties on the CO–PES fabric, homogeneous print quality was obtained in Figure 9.

### 3.3. Electrical Properties of Printed Surfaces

Consistent electrical conduction is one of the most critical requirements for an effective antenna performance because it impacts the return loss and radiation efficiency. These material qualities played an essential role in producing antenna patterns regardless of the applied printing techniques. The printed surface must be thick enough to assure good conductivity and low ohmic losses, usually achieved by integrating metallic or other particles as the conductive medium. However, using metallic inks is not so cost-effective and limits the use in textile applications by bringing additional costs and oxidation issues. The thickness associated with the number of printed layers determines the conductivity and fabric properties, including comfort, weight, and outcomes, including cost and process time. In this study, we examined the sheet resistance and electrical conductivity variation as a function of the printed number of layers for both PICA and LOOP patterns.

As discussed in detail here, a single layer of CNTs/CNC ink was not sufficient to achieve the required low sheet resistivity; thus, stacking multilayers together and doping them could be suggested to achieve lower sheet resistance. Figure 10a presents the sheet resistance of CNTs/CNC 1:1 and CNTs/CNC—1.5:1.5 printed PICA pattern on CO fabrics decreased by ~95% from 405 ± 58 Ω/sq to 22 ± 1.9 Ω/sq and from 233 ± 40.2 Ω/sq to 15 ± 2.0 Ω/sq with an increase in the number of layers from 1 to 5, respectively. The rule of thumb for decreasing extrinsic sheet resistance values as low as 15 ± 2.0 Ω/sq is to increase the thickness of CNTs using layer-by-layer piling, and doping the CNTs can allow the extrinsic sheet resistance values to be reduced to as low as 15 ± 2.0 Ω/sq. After the first printed layer, the decrease in sheet resistivity for both PICA and LOOP designs was attributed to the less void content between the conductive layer and the fabric surface and higher CNTs deposition [79,80]. After applying the second layer, when the interconnecting CNTs formed a conductive pathway for electrons that promoted a large number of electrons flowing, lower sheet resistance was recorded. The effect of the number of layers increasing, resulting in a decrease in sheet resistance, on the conduction mechanism of thin films was also noted [81,82,83].

Figure 10c demonstrated the sheet resistance changes of LOOP design depending on the number of layers. The same trend was observed in the LOOP design; after a big jump in conductivity level, slight increase in conductivity was noted by increasing print layers.

Utilizing CNTs/CNC—1:1 and CNTs/CNC—0.5:1 for LOOP patterns with 5 printed layers reduced the sheet resistance from 11.5 ± 1.5 kΩ/sq to 176 ± 37 Ω/sq and from 3.7 ± 2.7 kΩ/sq to 326 ± 78 Ω/sq, respectively. The penetration of ink into the fabric was affected by a range of material factors, such as the surface tension, viscosity of the ink, the volatility of ink, and the wettability of ink to the fabric [84]. All these parameters also impacted on ink deposition and surface roughness [85]. Thus, the difference in sheet resistance between the LOOP and PICA patterns could be attributed to the nature of the fabric surface and the low dispersion absorbency of CO–PES fabric compared to CO fabric.

The electrical conductivity was calculated following the equation where *σ*, *ρ*, *t* are the electrical conductivity, resistivity, and thickness of layers, respectively.
*ρ* = (*π*/*ln*2)(*V*/*I*)*t*,(4)
*σ* = 1/*ρ*,(5)

As shown in Figure 10b, the conductivity of printed PICA patterns increased with the number of layers from 12.3 ± 2.2 S/cm to 27.6 ± 2.4 S/cm up to the 4th layer and then slightly decreased to 20.4 ± 0.6 S/cm in CNTs/CNC—1:1. In CNTs/CNC—1.5:1.5 printed PICA pattern, the conductivity of printed designs was noted as 8.6 ± 2.0 S/cm at the first layer, then reached to *c.a* 20 ± 1.4 S/cm. For CNTs/CNC—1:1, surface conductivities measured on LOOP patterns were recorded as 0.40 ± 0.20 S/cm, 2.1 ± 1.2 S/cm, 3.5 ± 0.3 S/cm, and 5.5 ± 0.7 S/cm up to the 4th layer, and then reduced to 5.2 ± 0.2 S/cm at the fifth layer as shown in Figure 10d. The conductivity values obtained from the CNTs/CNC—0.5:1 were higher than the CNTs/CNC—1:1 dispersion that was 4.3 ± 0.7 S/cm, 5.4 ± 1.6 S/cm 7.1 ± 2.3 S/cm up to the 3rd layer respectively, and then decreased 6.2 ± 1.8 S/cm and 5.3 ± 1.1 S/cm in 4th and 5th layers, as shown in Figure 10d. The decrease in conductivity of both PICA and LOOP patterns after applying the 5th layer can be explained by the increment of thickness while the resistance remains relatively constant according to the formulation *σ = 1/ρ*. Liu et al. reported the electrical conductivity of a 15 wt.% carbon black-based ink printed sample as 2.15 × 10^4^ S/m [86]. As in our study, they reported that the sheet resistance of printed patterns is in the tendency to decrease with the increase in pattern thickness, which causes a reduction in the electrical conductivity. In addition, as seen in Figure 10b,d, the electrical conductivity values of the samples obtained from the ink with less CNTs ratio were higher for both PICA and LOOP patterns. It was ascribed to the fact that the ink containing less solid content created a thinner layer for LOOP and PICA patterns. As mentioned in Section 2.4.1 and Section 2.4.2, the average thickness of the CNTs/CNC—1:1 printed PICA pattern was 25 ± 2.9 μm, while the CNTs/CNC—1.5:1.5 printed PICA pattern had 35 ± 2.1 μm. CNTs/CNC—0.5:1 and CNTs/CNC—1:1 printed LOOP sample average thicknesses were 6.4 ± 0.7 μm and 14 ± 5.2 μm, respectively. Consequently, the thickness of samples prepared with low solid content inks was low and according to Equations (4) and (5), when the thickness decreases, the electrical conductivity increases. It is important to note that due to lower water absorption of the CO–PES fabric, the water based inks were not penetrated into the fabric thoroughly as in the CO fabric. Hence, the thickness of the printed layer was thinner than in CO fabric samples [87]. As a result, the low solid content samples which were CNTs/CNC—0.5:1 for LOOP and CNTs/CNC—1:1 for PICA pattern had higher electrical conductivity compared to high solid content inks which were CNTs/CNC—1:1 for LOOP and CNTs/CNC—1.5:1.5 for PICA.

### 3.4. Fabric Care: Washing and Ironing Tests

In the context of wearable electronics, apart from intrinsic material properties including electrical conductivity, fabric qualities have a crucial role in final application. In this study, the textiles antennas were tested in terms of washing and ironing stability. Figure 11 displays the deprivation in conductivity as a function of four wash cycles in 5 layers CNTs/CNC—1:1 and CNTs/CNC—1.5:1.5 printed PICA samples. Even though the washed samples still were able to conduct electrons, after four wash cycles, the surface resistivity of CNTs/CNC—1:1 increased from 5.9 ± 0.4 Ω to 52.9 ± 17.3 Ω. The same decrease in conductivity was also noted for CNTs/CNC—1.5:1.5, the resistivity was increased from 3.3 ± 0.5 Ω to 112 ± 39 Ω. This distress after each wash cycle can be attributed to accumulation of surfactants through the cracks on the surface, and damaging layer bonding and CNTs conductive pathway due to mechanical friction [80]. The deterioration of the printed layers causes an increase in its resistance to a value. Several studies reported a possible enhancement of electrical conductivity with heat post treatment that eliminates the surfactants [88]. Thus, between each wash cycle, the textile antennas were ironed, but there was no noticeable change noted. In the literature, the surfaces of such samples are usually coated with a polymer [17] or plastisol ink which has PVC particles in it as a plasticizer [89] that make the structure more durable in repeated wash. This study reports a polymer free materials approach without any coatings; the change in resistance after the first wash only increased from 5.9 ± 0.4 to 7.2 ± 0.6 Ω and from 3.2 ± 0.4 to 5.1 ± 1.9 Ω for CNTs/CNC—1:1 and CNTs/CNC—1.5:1.5, respectively.

### 3.5. Characterisation of Antenna Parameters

Their EM characteristics are analysed using simulated and measured data. Return loss (*S*_11_), in combination with far-field properties and efficiency values, demonstrates the responsiveness of these prototypes in wireless body area networks (WBAN) and wireless personal area networks (WPAN).

#### 3.5.1. Impedance Bandwidth

Figure 12 presents *S*_11_ computational and experimental results for the PICA design in both scenarios, off- and on-body settings. Experimental measurements were carried out for both concentrations of 1.0 wt.% and 1.5 wt.%. Results suggest the ultrawideband behaviour of the antenna with a bandwidth of 8.25 GHz and 8.5 GHz in the free-space scenario, for 1.0 wt.% and 1.5 wt.% cases, respectively; while for the on-body case, return losses less than 10 dB are slightly narrower in range, from 3.5–11.5 GHz at lower CNTs concentration and widened from 2.5 to 11.5 GHz by increased CNTs concentration. Measurement results do not possess the multi-frequency resonance characteristic due to the gradual impedance matching across the frequency band, so the resonance effect revealed a traveling wave antenna steadily. Nevertheless, the proposed antenna design and materials, whether free-space or on-body, cover a wide bandwidth range that fulfils the requirements of many wireless standards, including UWB communications [90,91].

Fabric-based devices are exposed to prolonged stresses of various kinds during their use and after-care, attaching themselves onto both irregularly shaped surfaces and the body [92]. To examine bending endurance, the antennas were placed on two foam cylinders (Rohacell) with *ε**_r_* close to 1 mimicking the air to consider the bending effect alone. Two cylinders of 70 and 150 mm radius were used to replicate a similar degree of conformity to that of lower and upper human limbs [7]. Figure 13 shows the simulation and measured PICA design results for the two cylinders, with a slight detuning frequency in each bending ratio. Higher bending ratio causes lower resonance length, and therefore, the resonance frequency increases as also noted in PICA designs in Figure 13.

Figure 14 shows the numerically computed and measured reflection coefficient of the wearable LOOP antenna. A slight shift towards lower frequencies is observed for both cases, with 2.41 GHz (simulated) and 2.39 GHz for 0.5 wt.% (CNTs/CNC—0.5:1), and 2.385 GHz 1 wt.% (CNTs/CNC—1:1) (measured). Lower return loss values towards lower frequencies are calculated in the case of on-body measurements.

The same approach as in the previous example has been followed for this model to investigate the antenna’s return loss performance under bending circumstances. Figure 15 depicts the effects of bending on the textile wearable loops prototypes when bent on a foam cylinder. In the case of the LOOP antenna designs, as depicted in Figure 15, we observed that the frequencies are shifted towards lower values. This phenomena is associated with its resonance frequency which relies on the perimeter of the loop. Furthermore, a slight reduction in resonance level resulting from the disturbance of current distribution while on bending mode was also noted [56].

#### 3.5.2. Radiation Pattern

In order to understand better the performance of the antenna, radiation pattern measurements were taken inside the anechoic chamber detailed in Section 2.9 to evaluate the radiation properties of the antenna in free space and on the human phantom. It defines the radiated power by an antenna as a function of the arrival angle observed in the antennas’ far field. Figure 16 shows the CST-simulated radiation patterns (black) of the PICA antenna, along with the measurements (red 1.0 wt.% CNTs/CNC—1:1 and blue 1.5 wt.% CNTs/CNC—1.5:1.5) in the off-body case. Three main frequencies along the operation frequency band were selected as 4, 7, and 10 GHz. In each frequency, electric field *E*-plane (φ = 90°) and magnetic field *H*-plane cuts (φ = 0°) were shown. The antenna showed an omnidirectional pattern; this was expected, due to its CPW design with no background plane to enhance the antenna’s directivity. At higher frequencies (7 and 10 GHz), the omnidirectional patterning was distorted due to the slight mismatch caused by the optimisation of antenna dimensions at 3 GHz. Measured values are well correlated with simulations with minimum variation and a slight ripple effect, where an omnidirectional pattern can be appreciated. These slight variations are associated with experimental deviations caused by the measurement setup and the flexible nature of the fabric antenna itself. The rotation generated physical tension on the flexible antenna and its SMA connector, thus, also caused these variations.

For on-body characterisation, the phantom previously described was employed in the test campaign. The 2D radiation pattern representation is depicted in Figure 17. The effect of the human body’s presence on the antenna’s performance can be seen, increasing the front-back ratio due to the lack of background plane in the design and because the body is a dissipative medium that absorbs the backward emitted energy. Figure 17a–f suggest the physical torso phantom absorbs less energy than the computed numerical models. This deviation in behaviour could be due to the complexity of reproducing the human body and its exact representation.

Figure 18 depicts the 2D view of the omnidirectional radiation pattern for the wearable loop antenna when simulating free-space conditions; it behaves the same as a short horizontal dipole antenna (red 0.5 wt.% CNTs/CNC—0.5:1 and blue 1 wt.% CNTs/CNC—1:1). As noted, the energy is radiated in an omnidirectional pattern with a distinctive toroid shape alongside with two-lobe opposite radiation directions that are separated 180° between each other. Figure 19 exhibits the on-body case for wearable applications; the robustness of the wearable loop antenna when close to human tissues needs to be maintained. As previously addressed, at these frequencies, the human body acts a reflector that is capable of enhancing the front–back ratio.

#### 3.5.3. Antenna Efficiency and Gain

Efficiency measurements were taken using a wheeler cap and following its method for the three selected frequencies. An accurate comparison is not possible due to the lack of references of CNTs screen printed antennas onto textiles within the microwave range. Values are lower than values previously reported in the literature for CNTs films on polyimide substrate [37,39]. Fabric substrates deal with an order of magnitude higher in terms of dissipation factor; therefore, reducing the efficiency of the whole antenna. The final parameters for the PICA model are listed in Table 6 and gain values for both ink formulations (simulated and measured) are noted in Table 7.

Regarding the LOOP design, its radiation efficiency simulated is 65%, while the measured value with the wheeler cap is ~10% at 2.4 GHz; despite this discrepancy, this outcome remains close to reported values of compact antennas for wearable sensor networks [93] onto flexible textile substrates. High-efficiency textile devices are a significant challenge within fabric-based electronics [94]. Nevertheless, the PICA with a CPW-fed printer demonstrated good coherency between the numerical estimation and measured data. For both ink formulations, the results suggested an operational bandwidth of at least 8 GHz (3.5–11.5 GHz) for both in free-space and on-phantom settings, ensuring the ultra-wideband mode for these prototypes. Two bending scenarios were considered to address some of the hurdles of a highly dynamic environment, in the context of the body. Numerical and experimental analyses were carried out to determine the impact of being near the human body on the antenna’s features such as *S*-parameters and radiation compared to the off-body scenario. While an omnidirectional radiation pattern was observed in the free-space scenario due to its co-planar structure, an increased front-to-back (*F*/*B*) was recorded in the on-body case due to the human body absorbing the energy emitted by the antenna. When loop designs were tested as an antenna, for prospect use in wearable and healthcare applications, *S*_11_ measured values slightly shifted towards lower frequencies, 20 and 25 MHz in CNTs/CNC—0.5:1 and CNTs/CNC—1:1. Simulated and measured return loss and far-field parameters for both scenarios, off- and on-body, were reasonably correlated. The antenna performance under bending for both PICA and LOOP designs revealed only a slight frequency detuning in the response that could be negligible to the UWB model, but it has to be considered for the resonance frequency of the LOOP. Low efficiency was noted within range for wireless communications. Thus, through material intervention, improving radiation efficiency could lead the way forward to tackle major challenges for CNTs and textile-based antennas.

Previous works reported CNTs compounds on different substrates such as paper, photo paper, or rubber [46], but the tunability and adaptability of textile substrates have not been fully explored [3]. Contrary to traditional metallic material solutions, the intrinsically flexible CNTs specimens promise a high degree of movement freedom and elongation with a moderate electrical conductivity. Manipulating different textiles substrates could enable full integration of technology into the fabrics, boosting the development of smart clothing. Nonetheless, translating nano-level superior properties of CNTs into the macro-scale still needs to be altered for efficient communication channels. It is still important to note that CNC played a critical role in ensuring good quality of sensory surfaces printing with a trade-off in electrical conductivity.

## 4. Conclusions

The electronics industry urges us to find sustainable material solutions that can replace some metallic components that are hardly resourced, recycled, or repurposed when coupled with other mediums such as natural or synthetic textiles. New electronic solutions should ground both on practical and environmentally friendly making and sustainable materials avoiding toxic substances. Combination of redundant CNC with a carbon allotrope, CNTs, can promise unique features such as adaptability, washability, and degradation to the electronic elements for sensing and communication in the context of smart wearables. This study is looking into new material led approaches for wearable communication. Within the purpose of understanding these new mediums and techniques, two CNTs/CNC antenna designs, such as LOOP and PICA, screen printed onto the flexible textile-based substrates, were studied. Through the successful formulation of water-based screen printing conductive inks with CNTs and CNC, we showed a facile approach that does not rely on inks with binder, solvent exchange, or requiring high-temperature heat post-processes annealing. Thus, on that note, these formulated inks can be applicable to other textiles printing processes by tailoring the solution viscosity and be used to work with a wide variety of flexible substrates including textiles with roughness. The work presented here strategically merged the new and nanoscopic forms of CNTs and cellulose, with a very established printing technique, screen-printing method to develop a new generation of textile-based wearable antennas. Inherent advantages of fabric features such as lightness, adaptability, and washability were noted along with the antenna measurements both on and off-body settings.

## Figures and Tables

**Figure 1 sensors-21-04934-f001:**
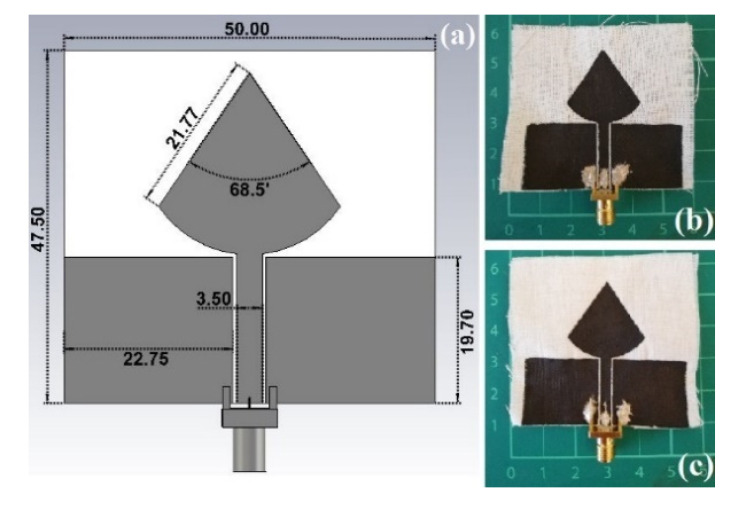
(**a**) PICA antenna design CAD model with the dimensions in mm; Prototyped models: (**b**) CNTs/CNC—1:1; (**c**) CNTs/CNC—1.5:1.5.

**Figure 2 sensors-21-04934-f002:**
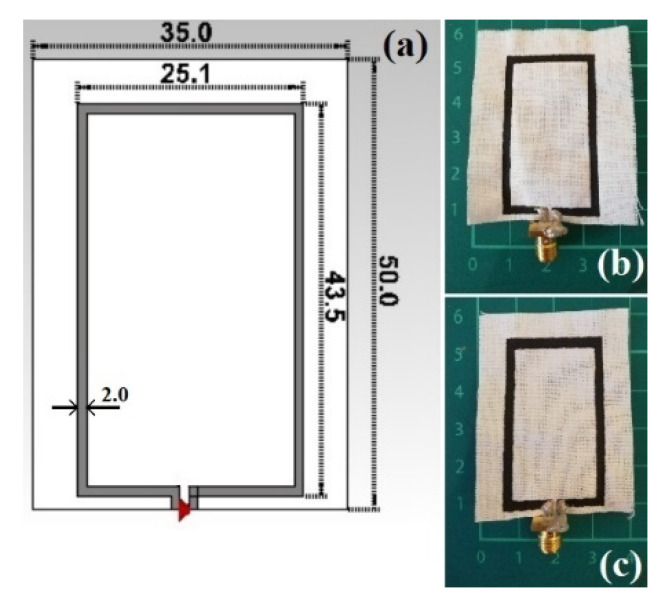
(**a**) Top view of the modelled textile loop printed antenna along with its dimensions in mm; Prototyped models: (**b**) CNTs/CNC—0.5:1; (**c**) CNTs/CNC—1:1.

**Figure 3 sensors-21-04934-f003:**
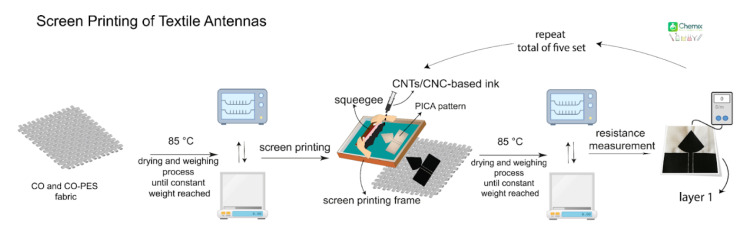
Schematics of screen printing of PICA and LOOP patterned textile antennas from CNTs/CNC formulations.

**Figure 4 sensors-21-04934-f004:**
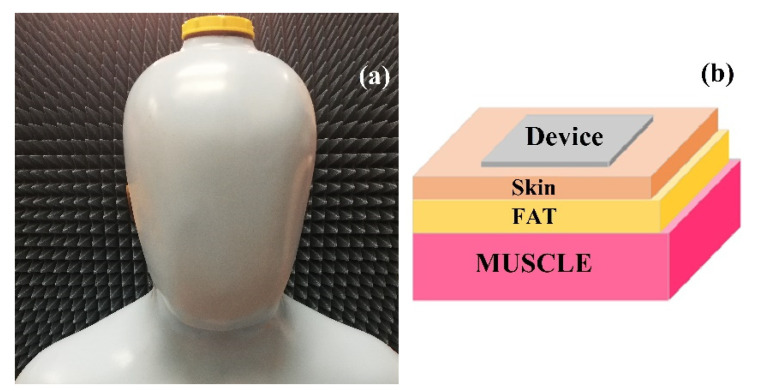
(**a**) A real image of the human torso phantom model; (**b**) A 3D view of a layered numerical representation of human tissue.

**Figure 5 sensors-21-04934-f005:**
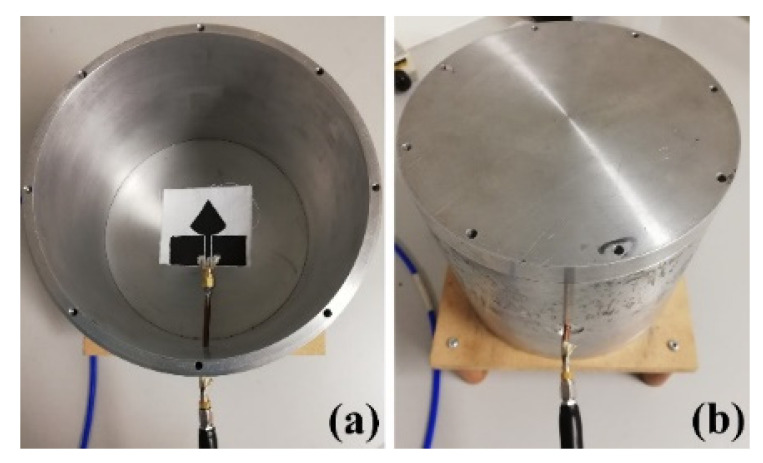
Laboratory model of a cylindrical wheeler cap: (**a**) Cap-off; (**b**) Cap-on.

**Figure 6 sensors-21-04934-f006:**
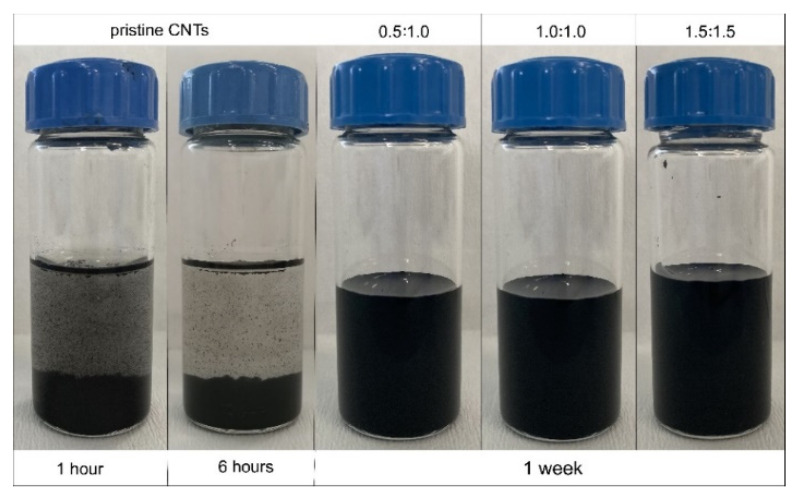
Dispersion and stability state of the CNTs in water with and without CNC. The ink compositions are represented from CNTs/CNC—0.5:1; CNTs/CNC—1:1; CNTs/CNC—1.5:1.5, left to right.

**Figure 7 sensors-21-04934-f007:**
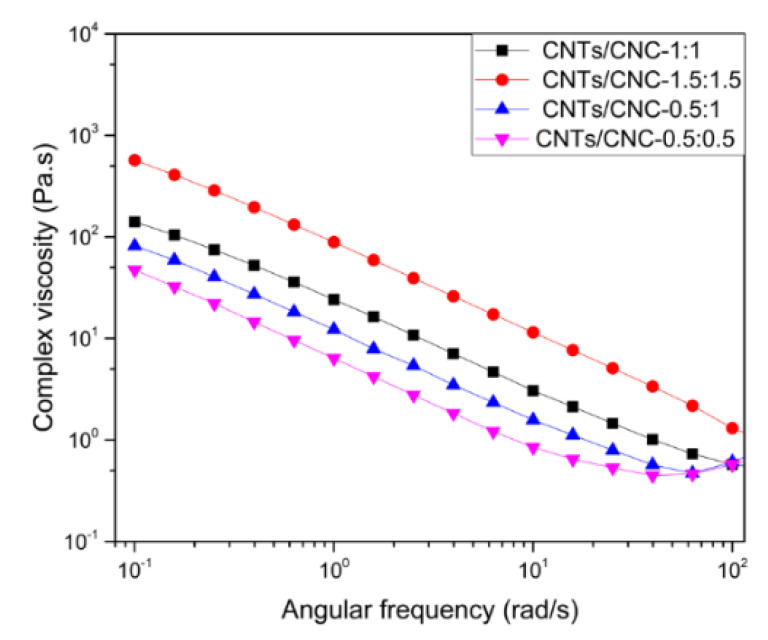
Complex viscosity of CNTs/CNC-based inks versus angular frequency.

**Figure 8 sensors-21-04934-f008:**
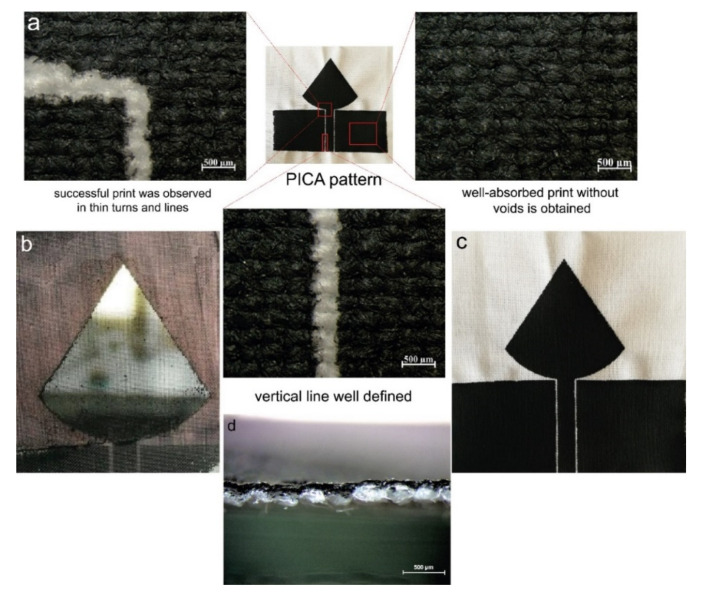
(**a**) Optic microscope images of CNTs/CNC—1.5:1.5 ink printed PICA pattern on CO fabric from different points; (**b**) A real picture of clogging meshes after using CNTs/CNC—1.5:1.5 ink; (**c**) Real image of CNTs/CNC—1:1 ink printed PICA pattern on CO fabric; (**d**) Cross-sectional optic microscope images of CNTs/CNC—1.5:1.5 ink printed CO fabric.

**Figure 9 sensors-21-04934-f009:**
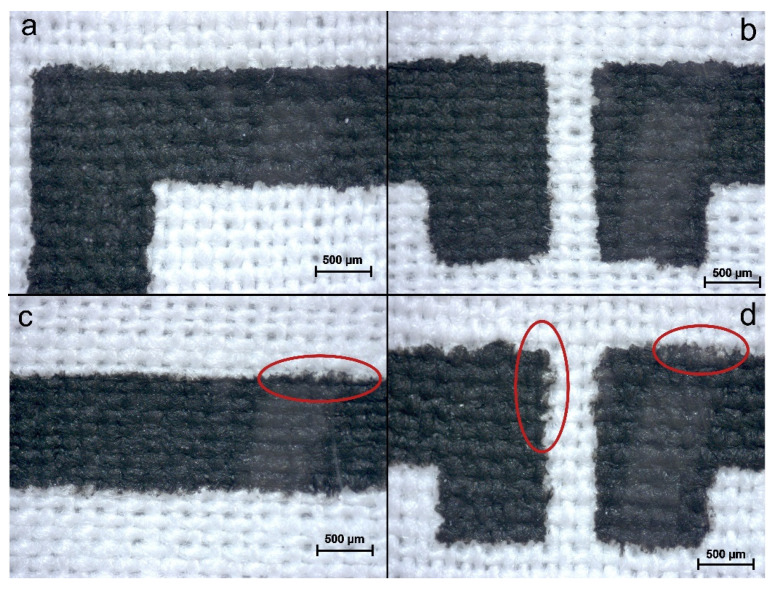
LOOP pattern images: (**a**,**b**) CNTs/CNC—1:1 ink printed on CO–PES fabric; (**c**,**d**) CNTs/CNC—0.5:1 printed CO–PES fabric, where red circles pointed out irregularities associated with ink concentration.

**Figure 10 sensors-21-04934-f010:**
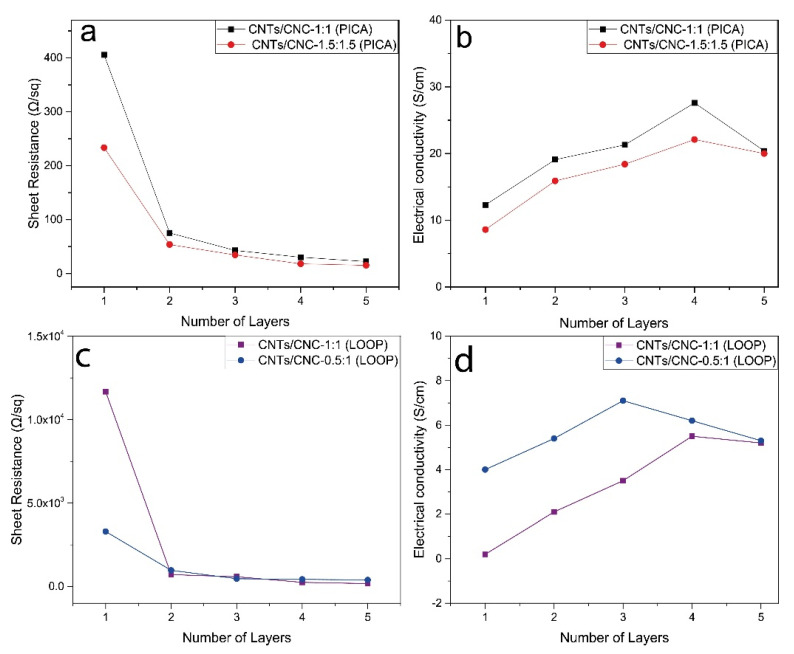
Resistance and electrical conductivity of CNTs/CNC printed samples concerning the number of printed layers; (**a**,**b**) CNTs/CNC—1:1 and CNTs/CNC—1.5:1.5 printed PICA designs, (**c**,**d**) CNTs/CNC—0.5:1 and CNTs/CNC—1:1 printed LOOP antenna as a function of deposited conductive layers.

**Figure 11 sensors-21-04934-f011:**
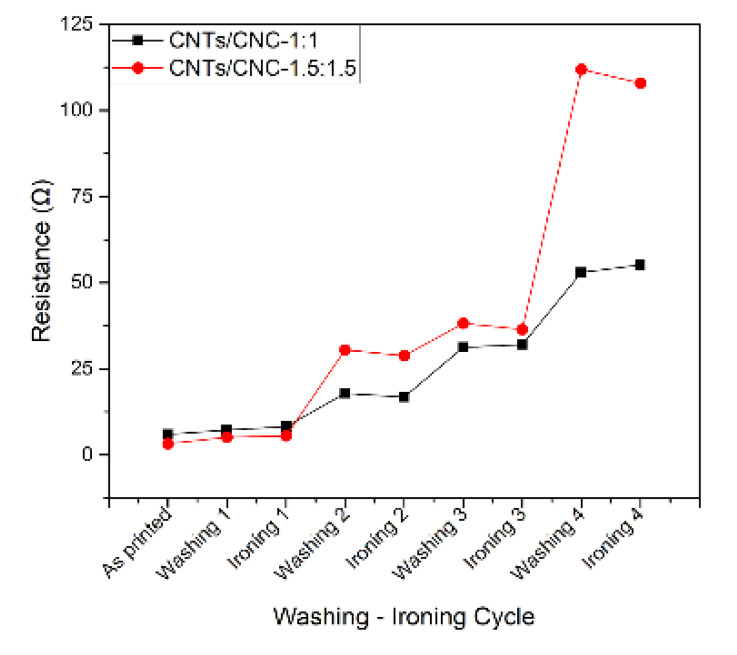
Resistance change after 4 washing and ironing cycles; refers to 5 layers CNTs/CNC—1:1 and CNTs/CNC—1.5:1.5 inks screen printed samples.

**Figure 12 sensors-21-04934-f012:**
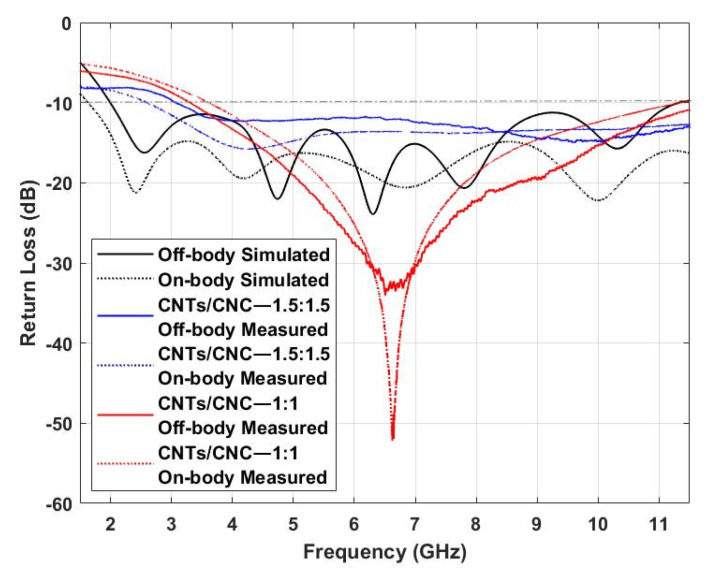
Simulated vs. Measured *S_11_* of CNTs/CNC PICA antennas in off- and on-body settings.

**Figure 13 sensors-21-04934-f013:**
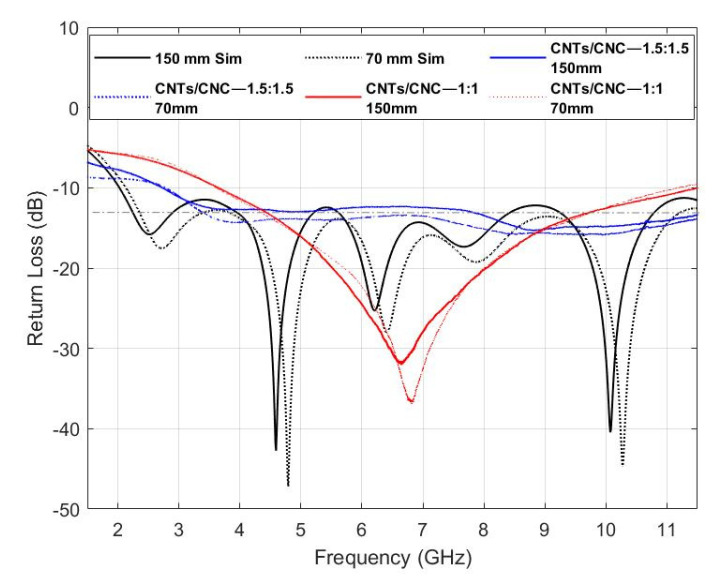
Simulated vs. Measured *S_11_* of CNTs/CNC PICA antennas under 70 and 150 mm bending scenarios.

**Figure 14 sensors-21-04934-f014:**
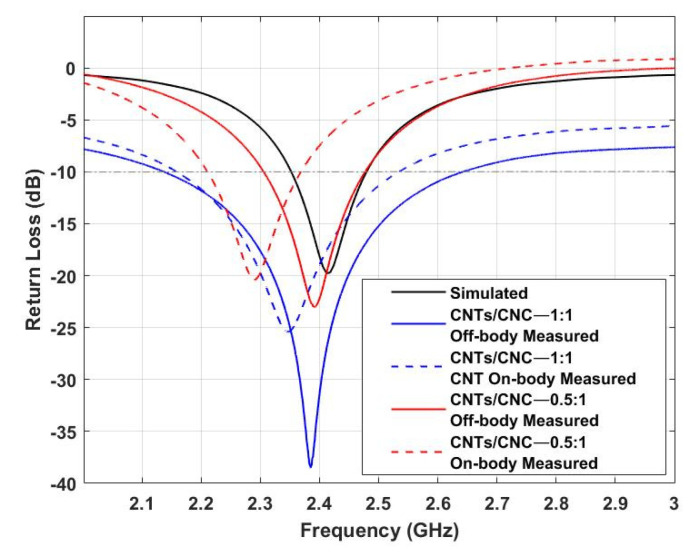
Simulated vs. Measured *S_11_* of CNTs/CNC LOOP antennas in off- and on-body settings.

**Figure 15 sensors-21-04934-f015:**
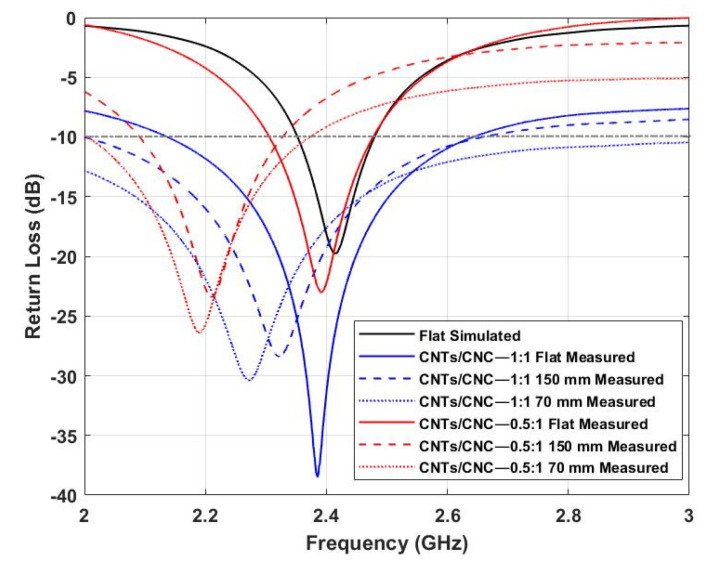
Simulated vs. Measured *S_11_* of CNTs/CNC LOOP antennas under 70 and 150 mm bending scenarios.

**Figure 16 sensors-21-04934-f016:**
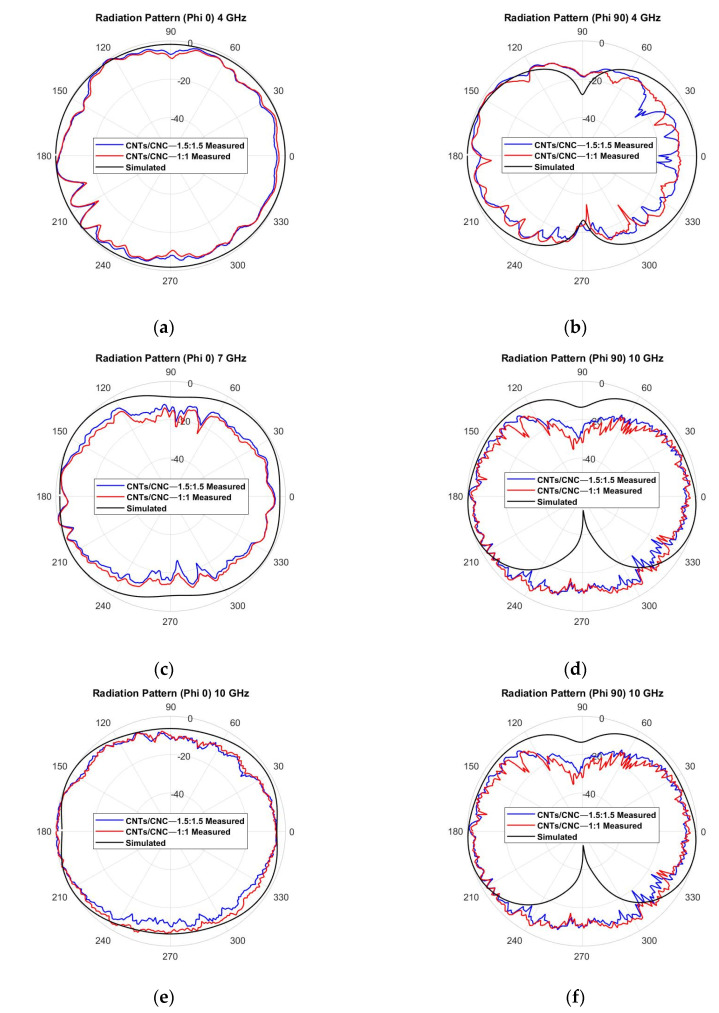
Measured vs. simulated radiation pattern of the graphene-based textile PICA antenna in free space, *E*-plane cut at φ = 90° and *H*-plane cut at φ = 0°, for the three main frequencies: (**a**,**b**) 4 GHz; (**c**,**d**) 7 GHz; (**e**,**f**) 10 GHz.

**Figure 17 sensors-21-04934-f017:**
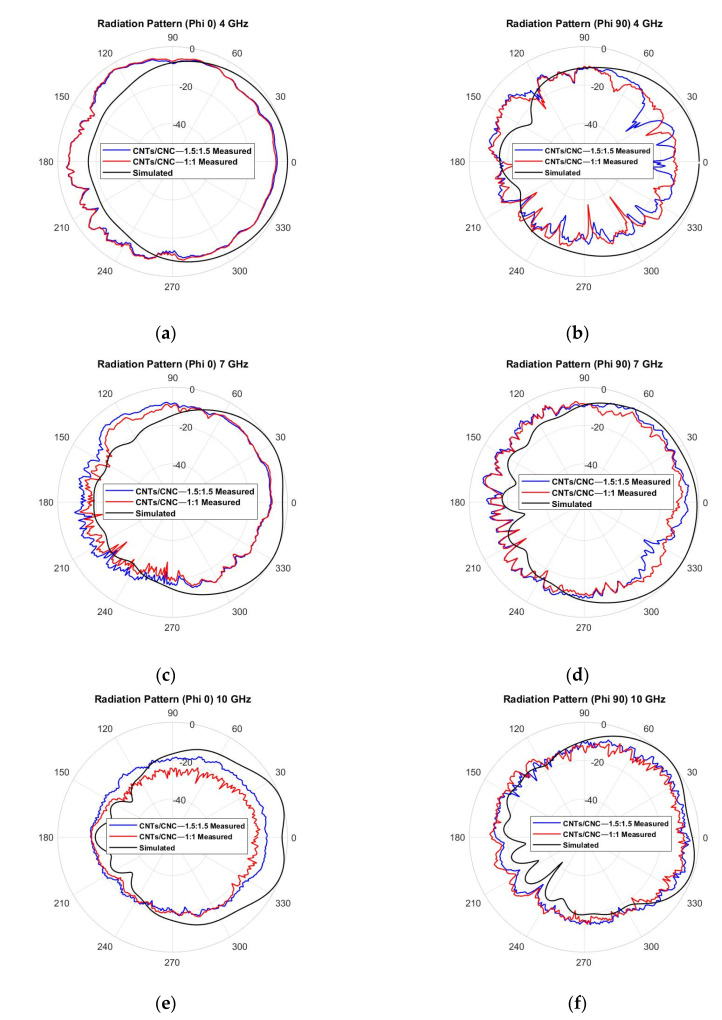
Measured vs. simulated radiation pattern of the graphene-based textile PICA antenna on the phantom, *E*-plane cut at φ = 90° and *H*-plane cut at φ = 0°, for the three main frequencies: (**a**,**b**) 4 GHz; (**c**,**d**) 7 GHz; (**e**,**f**) 10 GHz.

**Figure 18 sensors-21-04934-f018:**
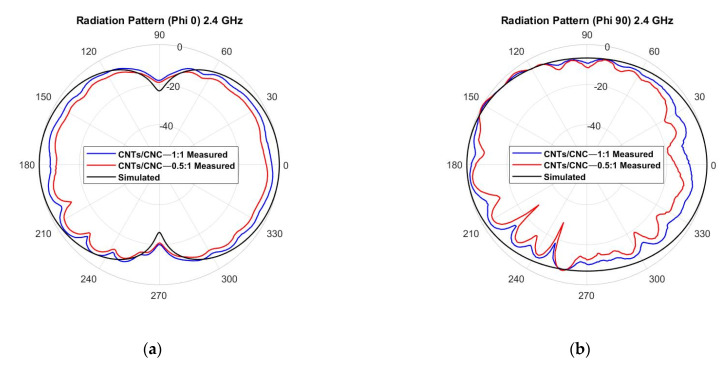
Measured vs. simulated radiation pattern of the graphene-based textile LOOP antenna in free space, *E*-plane cut at φ = 90° and *H*-plane cut at φ = 0°.

**Figure 19 sensors-21-04934-f019:**
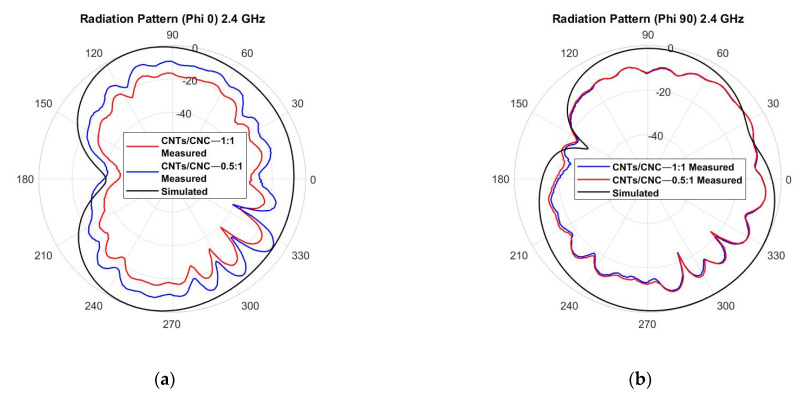
Measured vs. simulated radiation pattern of the graphene-based textile LOOP antenna on the phantom, *E*-plane cut at φ = 90° and *H*-plane cut at φ = 0°.

**Table 1 sensors-21-04934-t001:** CNTs/CNC ink compositions with related sample codes and printed antenna designs.

Sample Codes	CNTs/CNC Ink Compositions	Sonication (Hour)	Antenna Designs
CNTs/CNC—0.5:0.5	0.5 wt.% CNTs and 0.5 wt.% CNC	1.0	PICA and LOOP
CNTs/CNC—1:1	1 wt.% CNTs and 1 wt.% CNC	1.0	PICA and LOOP
CNTs/CNC—1.5:1.5	1.5 wt.% CNTs and 1.5 wt.% CNC	1.5	PICA
CNTs/CNC—0.5:1	0.5 wt.% CNTs and 1 wt.% CNC	1.0	LOOP

**Table 2 sensors-21-04934-t002:** Parameters of screen printing on CO fabric using CNTs/CNC—1:1 and CNTs/CNC—1.5:1.5.

Number of Layer	Volume of Dispersion (mL)	Number of Squeegee Movement
1	0.5	5
2	0.4	2
3	0.4	2
4	0.4	2
5	0.4	2

**Table 3 sensors-21-04934-t003:** Parameters of screen printing on CO–PES fabric using CNTs/CNC—1:1 and CNTs/CNC—0.5:1.

Number of Layers	The volume of Dispersion (mL)	Number of Squeegee Movement
1	0.4	5
2	0.4	4
3	0.3	4
4	0.3	4
5	0.3	4

**Table 4 sensors-21-04934-t004:** Electromagnetic properties of human tissues tested at 2.45 GHz [61].

Tissue	*ε_r_*	*tanδ*	*σ* (S/m)	*ρ* (Ω·m)
Dry Skin	38.007	0.283	1.464	0.683
Fat	5.280	0.145	0.105	9.568
Muscle	52.729	0.242	1.739	0.575

**Table 5 sensors-21-04934-t005:** Complex viscosity values at different angular frequencies.

Complex Viscosity (Pa·s)				
Ink Code	0.1 (rad/s)	1.0 (rad/s)	10 (rad/s)	100 (rad/s)
CNTs/CNC—0.5:0.5	47.3 ± 0.2	6.4 ± 0.2	0.84 ± 0.3	0.6 ± 0.1
CNTs/CNC—1:1	141 ± 6.8	24.1 ± 8.5	3.05 ± 1.2	0.6 ± 0.2
CNTs/CNC—1.5:1.5	783 ± 7.1	132 ± 1.0	17.3 ± 0.2	2.2 ± 0.3
CNTs/CNC—0.5:1	81.6 ± 6.9	12.2 ± 0.7	1.6 ± 0.1	0.6 ± 0.1

**Table 6 sensors-21-04934-t006:** PICA efficiency simulations and measurements at 4, 7, and 10 GHz for both ink formulations.

Frequency (GHz)			4	7	10
Efficiency (%)	Simulated	CNTs/CNC—1:1	26	37	47
		CNTs/CNC—1.5:1.5	27	39	50
	Measured	CNTs/CNC—1:1	20	25	34
		CNTs/CNC—1.5:1.5	22	27	40

**Table 7 sensors-21-04934-t007:** PICA gain simulations and measurements at 4, 7, and 10 GHz for both ink formulations.

Frequency (GHz)			4	7	10
Efficiency (%)	Simulated	CNTs/CNC—1:1	−6.28	−3.82	−1.96
		CNTs/CNC—1.5:1.5	−6.10	−3.59	−1.70
	Measured	CNTs/CNC—1:1	−7.40	−5.52	−3.37
		CNTs/CNC—1.5:1.5	−7.00	−5.19	−2.60

## Data Availability

The data presented in this study are available on request from the corresponding author.

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
