# Peer review of "Screen Printing Carbon Nanotubes Textiles Antennas for Smart Wearables"

_sensors, 2021, doi:10.3390/s21144934_

Round 1

Reviewer 1 Report

The work generally looks good. From my point of view, due the extensive introduction and references, I think a more detailed explanation in the characterization of the antenna parameters and radiation pattern should be done. This should be done with reference not only to the direct application discussed in the paper but also to other specific properties.

What means the shift between the experimental data and simulation (fig16 to fig 18)?

In the radiation pattern, what means the change in the simulated shape? Why this happens?

From my knowledge, 20 washes should be done to the textiles. Could you predict what happen to the resistivity after these 20 cycles?

In line 514, the sheet resistance is 2815±1807 ohms/sq. Is the error value correct?

Along the text appear %wt and wt.%, please be coherent. The authors should be use wt.%

Line 255, 256, 270. Etc. Please be careful with decimal places and significant figures (numbers).

Author Response

We want to thank the reviewer for the time spent on this manuscript and the comments and suggestions provided. We attach a Word file with a point-to-point response to the reviewer's comments.

Reviewer 2 Report

"Screen Printing Carbon Nano-Tubes Textiles Antennas for Smart Wearables" is very actual paper and addresses important issues. It is generally well written. Some thing can be improved however to enhance the clarity. Sonication times could be added to table 1. Figure 3 could be enhanced, by clearly indication repetition, i.e. feedback would be in order on the right side. Also, font size should be improved. Tables S1 and S2 could be near the actual text describing them. Rheological aspects are presented in telegram style (2.5, 3.2), and could be more detailed. Confusing usage of angular velocity versus frequency needs clearing. Figure 13 is missing. Figure 4 text should be more readable. Figures 14, 15, 16, 17, 18, 19 and 20 are too small to be readable in printed form. Overall, it is very interesting work.

I would rather focus on sentences about "Rheological aspects" and "angular velocity", which did get also adequate response. Generally, review follows exactly your suggestions:

> • Consider starting with a short summary of the manuscript explaining
> what the study is about

"Screen Printing Carbon Nano-Tubes Textiles Antennas for Smart Wearables" is very actual paper and addresses important issues. It is generally well written."

> • Then, explain each of the issues found that need to be addressed.
> Divide the list into major issues and minor issues.

"Some things can be improved however to enhance the clarity."

> • Major issues might include problems with the study’s methodology,
> techniques, analyses, missing controls or other serious flaws.

Empty, because not really present. At least according to my understanding of the “major issue”, as I also indicated "Accept after minor revision (corrections to minor methodological errors and text editing)". The reviews in similar cases tend typically be rather short.
It is debatable of course, nevertheless most of the reviewers seem to agree with me here, sometimes in much fewer words. In fact, Reviewer 4 has only the following sentence:

"Variables should be written in Italic. Some subscripts are missing or should be corrected, like subscript "r" of relative permittivity "εr" in table 2 and in the lines 306, 315 and 325."

> • Minor issues might include tables or figures that are difficult to
> read, parts that need more explanation, and suggestions to delete
> unnecessary text.

"- Sonication times could be added to table 1.
- Figure 3 could be enhanced, by clearly indication repetition, i.e. feedback would be in order on the right side. Also, font size should be improved.
- Tables S1 and S2 could be near the actual text describing them.
- Rheological aspects are presented in telegram style (2.5, 3.2), and could be more detailed.
- Confusing usage of angular velocity versus frequency needs clearing.
- Figure 13 is missing. Figure 4 text should be more readable.
- Figures 14, 15, 16, 17, 18, 19 and 20 are too small to be readable in printed form."

And finally:

"Overall, it is very interesting work."

And I still consider it to be the case according to my understanding, at least when "methodology of a study" and "scientific content of the article" is concerned. Things can always be enhanced, but the current case seemed to be sufficiently covered. 

Author Response

(The authors gave the same response as above.)

Reviewer 3 Report

1# In Introduction, some more recent published papers should be cited to emphasize the importance of printed electronics for conductive (nano)materials, such as https://doi.org/10.1002/admt.201800546, https://doi.org/10.1021/acsomega.1c00638. Moreover, in addition to printing metal nanoparticles (Ref. 7 in the manuscript), other metal materials can also be printed or deposited as conductive patterns, some papers can also be cited, such as https://doi.org/10.1038/ncomms8461, https://doi.org/10.1002/adma.201600772, https://doi.org/10.1021/acsanm.8b00830.

2# In Figure 2a, the width of the pattern should be marked (2 mm?). Moreover, it seems that the patterns in Figure 1c (2c) is bigger than that in Figure 1b (2b). Why? more solid concentrations lead to wider patterns?

3# Some figures can be moved as supporting information. For example, Figure 8 just showed some initial results (the printing results are bad as the spacing between the printed patterns is also filled with the ink (the bottom part of Figure 8a)) and can be removed and added to supporting information.

Author Response

(The authors gave the same response as above.)

Reviewer 4 Report

Variables should be written in Italic. Some subscripts are missing or should be corrected, like subscript "r" of relative permittivity "εr" in table 2 and in the lines 306, 315 and 325.

Author Response

(The authors gave the same response as above.)

Round 2

Reviewer 1 Report

The authors present the revised version of their paper. A major effort was done to answer the reviewer's questions. Some answers could be discussed, but I will only focus my remarks on the revised version and all of them were discussed and improved.